# Epigenetic Mechanisms in Parenchymal Lung Diseases: Bystanders or Therapeutic Targets?

**DOI:** 10.3390/ijms23010546

**Published:** 2022-01-04

**Authors:** Edibe Avci, Pouya Sarvari, Rajkumar Savai, Werner Seeger, Soni S. Pullamsetti

**Affiliations:** 1Department of Lung Development and Remodeling, Max-Planck Institute for Heart and Lung Research, Parkstrasse 1, 61231 Bad Nauheim, Germany; Edibe.Avci@mpi-bn.mpg.de (E.A.); Pouya.Sarvari@mpi-bn.mpg.de (P.S.); Rajkumar.Savai@mpi-bn.mpg.de (R.S.); Werner.Seeger@innere.med.uni-giessen.de (W.S.); 2Department of Internal Medicine, Justus Liebig University, 35392 Giessen, Germany; 3Institute for Lung Health (ILH), Justus Liebig University, 35392 Giessen, Germany

**Keywords:** IPF, COPD, DNA methylation, histone modifications, microRNA, epigenetic editing, gene transfer

## Abstract

Epigenetic responses due to environmental changes alter chromatin structure, which in turn modifies the phenotype, gene expression profile, and activity of each cell type that has a role in the pathophysiology of a disease. Pulmonary diseases are one of the major causes of death in the world, including lung cancer, idiopathic pulmonary fibrosis (IPF), chronic obstructive pulmonary disease (COPD), pulmonary hypertension (PH), lung tuberculosis, pulmonary embolism, and asthma. Several lines of evidence indicate that epigenetic modifications may be one of the main factors to explain the increasing incidence and prevalence of lung diseases including IPF and COPD. Interestingly, isolated fibroblasts and smooth muscle cells from patients with pulmonary diseases such as IPF and PH that were cultured ex vivo maintained the disease phenotype. The cells often show a hyper-proliferative, apoptosis-resistant phenotype with increased expression of extracellular matrix (ECM) and activated focal adhesions suggesting the presence of an epigenetically imprinted phenotype. Moreover, many abnormalities observed in molecular processes in IPF patients are shown to be epigenetically regulated, such as innate immunity, cellular senescence, and apoptotic cell death. DNA methylation, histone modification, and microRNA regulation constitute the most common epigenetic modification mechanisms.

## 1. Pathogenesis of IPF and COPD

Idiopathic pulmonary fibrosis (IPF) is a chronic and progressive disease of the lung with unknown etiology characterized by the excessive extracellular matrix (ECM) formation and remodeling of the lung architecture, and eventually death. According to the World Health Organization (WHO) mortality database, the mean age of the mortality rates from IPF across Europe in 2001–2013 were 3.75 per 100,000 for males and 1.50 per 100,000 for females [1]. The median survival among 65-year-old or older adults with IPF in the United States was 3.8 years [2]. The core pathogenesis of IPF seems to arise from the dynamic interplay between genetic, epigenetic, and environmental factors causing irritation, damage, senescence, and apoptosis of alveolar epithelial cells leading to the production of profibrotic factors responsible for activation of fibroblasts and their trans-differentiation into myofibroblasts [3,4,5,6]. Studies on the pathogenesis of the disease have mostly focused on the mechanisms that regulate proliferation, activation, and differentiation of alpha-smooth muscle actin (α-SMA) producing myofibroblasts. However, there is a need for an integrated and comprehensive understanding of the molecular mechanisms of IPF in a cell type-specific manner to develop more efficient therapeutics for this complex disease.

IPF is recognized as neoproliferative disorder of the lung and has been reported to have similarities and links to cancer biology. Epigenetic and genetic alterations, abnormal expression of microRNAs, gene mutations, activation of similar signaling pathways, apoptotic resistance, myofibroblast origin and behavior, altered cellular communications, intracellular signaling pathways, and unknown etiology are among various contributing pathogenic mechanisms common in both of these fatal disorders [7,8] (Figure 1). It is worth mentioning that these genetic and epigenetic mechanisms usually lead to aberrant activation of the epithelial cells in the first place, which produce the factors participating in the activation of the fibroblastic/myofibroblastic foci and the activation of a fibrotic response [9]. Moreover, the pathogenesis of other pulmonary diseases such as asthma and chronic obstructive pulmonary disease (COPD) has been already linked to epigenetics [10,11].

Chronic obstructive pulmonary disease (COPD) is a cigarette smoke- and aging-related progressive disorder, characterized by lung inflammation, emphysema formation, abnormal tissue repair, and alveolar destruction (Figure 2). Decline in lung function and airflow obstruction as characteristic of COPD is mainly connected to remodeling of small airways [12,13], which eventually leads to destruction of lung parenchyma. Airway wall thickening occurs progressively along with increased severity of the disease in patients with COPD [14].

COPD is the third leading cause of fatality worldwide, and the prevalence continued to increase by 2019, according to WHO records (WHO Global Health Estimates, 2019). To date, molecular characterization related to COPD has dominantly focused on oxidant–antioxidant balance, elastase–anti elastase hypothesis, chronic inflammation, apoptosis, and aberrant repair. Cigarette smoking is the main risk factor as well as other toxic inhalant and gases for developing COPD and inflammation [15,16]. Over 4700 chemical compounds, 73% of which have been identified as carcinogenic by IARC (International Agency for Research on Cancer), and a large number of free radicals activate transcription factors such as Nuclear factor kappa B NF-κB, leading to increased pro-inflammatory cytokine levels in COPD patients. Increased levels of reactive oxygen species (ROS) in lung airways and blood samples is a hallmark of impaired oxidative stress in COPD patients [17,18,19]. The damage due to generated free radicals/oxidants eventually results in alveolar epithelial injury, inactivation of pulmonary surfactants and anti-proteases, membrane lipid peroxidation, remodeling of blood vessels as well as extracellular matrix, dysregulated mitochondrial function, apoptosis, and lastly inflammation [20]. Inflammation itself is a major influencer of the ROS response in the cells [21]. Infiltrating cells such as leukocytes, macrophages, dendritic cells, and natural killer cells damage tissue by activating a persistent immune response due to reduced epithelial barrier integrity. Collectively, dysfunction of alveolar and bronchial epithelial cells, emphysema formation in gas exchange surface area in the lung, and severe tissue remodeling result in progressive airway wall thickening in COPD [22].

Currently, there is no effective clinical treatment available. Over the past two decades, anti-inflammatory inhaled corticosteroids in combination with bronchodilators have been the main therapeutic strategy to improve the health status of patients with COPD, and subsequently their quality of life [11,23]. However, large numbers of COPD patients are resistant to corticosteroids. Since COPD displays different phenotypes, new therapeutic approaches for personalized medicine are urgently requisite. In recent years, epidemiological studies indicate that up to 40% of all COPD patients are smoke-free yet suffer from the disease [24]. As COPD is not a monogenic disease and genetic contribution differs, epigenetic axis to COPD should be considered carefully. Thus, elucidating the epigenetic mechanisms underlying parenchymal remodeling in COPD and IPF are highly relevant to discover the novel treatments and regimes for these deadly diseases.

This review summarizes the dysregulation of various epigenetic mechanisms (DNA methylation, histone modifications, and non-coding RNAs) and their impact on pulmonary fibrosis and COPD, as well as the elaborate in vitro and in vivo studies that brought the crucial role of epigenetic mechanisms in disease pathogenesis to light. We also discuss the current preclinical status of pan- and isoform-selective histone deacetylase (HDAC) inhibitors, DNA methylation, and microRNA modulators and propose new research areas that may facilitate locus-specific epigenome editing as a novel therapeutic strategy for IPF and COPD.

## 2. Epigenetic Dysregulation in IPF and COPD

Epigenetics generally refers to heritable changes in gene expression that occur independent of DNA sequence. Epigenetic alterations can be subdivided into three main classes: DNA methylation, post-translational histone modifications, and non-coding RNAs. These epigenetic marks are induced by environmental factors, diet, accumulated somatic mutations over aging, and a variety of diseases [25,26]. Collectively, studies have emerged with evidence showing that epigenetic mechanisms impact on phenotypic changes in chronic lung diseases. Thus, epigenetics enlightens and provides a comprehensive link for genotype–phenotype correlation.

### 2.1. DNA Methylation in Lung Fibrosis

Methylation of cytosine residues within the CpG islands of gene promoters is known to block RNA polymerase complex and therefore suppresses gene expression. DNA methylation is shown to associate with altered expression of genes and pathways important to lung diseases such as IPF [27,28]. In addition, DNA hypermethylation at promoter regions is considered a crucial contributing factor for downregulation of the IPF suppressor genes such as *THY1* (*Thy-1 antigen*), *CAV1* (*Caveolin 1*), *PTEN* (*Phosphatase and tensin homolog*), and *PTGER2* (*Prostaglandin E receptor 2*), which are known to regulate important cellular processes. Moreover, MUC (mucins) are among key proteins involved in the regulation of cell growth and tissue remodeling processes whose genetic and epigenetic deregulation is associated with lung diseases such as IPF [29] and COPD [30]. Interestingly, it was shown that the hypermethylation of the *MUC5* promoter region is associated with IPF and that there is a mutation within the same region surrounding the *FOXA2* (Forkhead box A2) binding motif. This mutation can change the binding affinity for other transcription factors affecting the expression of *MUC5*. Genome-wide methylation profiling studies in both lung tissue [27,31] and isolated primary fibroblasts [32] identified extensive DNA methylation changes of patients with IPF compared with controls with substantial effect of these methylation changes on gene expression. DNA methyltransferases (DNMTs) are a family of enzymes responsible for maintaining DNA methylation. Recent studies show that there is an increased expression of DNMTs specifically for DNMT3a and DNMT3b, but no significant changes were observed in levels of DNMT1 in IPF lung tissues [24]. In addition, another study showed that application of *TGF-β1* (*Transforming growth factor beta 1*), a major contributing factor in IPF, increased the protein levels of DNMT1 and DNMT3a in lung fibroblasts without altering their mRNA expression by distinct post-transcriptional mechanisms [33]. Production of DNMT3a was increased by TGF-β1 via an increase in its protein synthesis and translation. By contrast, TGF-β1 inactivates glycogen synthase kinase-3β that causes inhibition of DNMT1 ubiquitination and its proteasomal degradation in lung fibroblasts. These studies suggest that DNA methylation is a crucial factor in the pathogenesis of IPF and that targeting DNMTs should be applied with caution in an isoform-specific and cell-specific manner.

### 2.2. DNA Methylation in COPD

DNA methylation is commonly linked to gene repression even though its effect is dependent on the location and cell type [34] and regulates the important pathways in COPD [35,36]. Recent studies have shown that a great number of CpG methylation sites are associated with both occupancy of the disease and severity of the symptoms in COPD patients [37]. The change in DNA methylation patterns of the promoters of pro-inflammatory genes has been revealed in alveolar epithelial cells and alveolar macrophages of patients with COPD [38]. DNA methylation is a key factor for developing COPD pathogenesis since this epigenetic mark is largely altered by aging and cigarette smoke through triggering severe inflammatory response, at last leading to the disease development [39].

The connection between DNA methylation pattern and cigarette smoke exposure with or without COPD has widely been analyzed through epigenome-wide association studies (EWASs). The largest EWAS held in 2013 by Zeilinger identified 187 differentially methylated CpG sites between non-smokers and smokers, and the group reported that reduced methylation levels were linked to cigarette smoking actively [40]. The cigarette smoke exposure has been shown to alter methylation pattern and in the context of the termination of tobacco consuming allows restoration to the methylation pattern of the non-smokers [41]. In addition, aberrant DNA methylation status of *GATA4* (*GATA binding protein 4*) and *p16* promoters obtained from sputum samples has been connected with decline in lung function in COPD [42]. Another study using small airway epithelial cells discovered the differences of 1260 methylated CpG sites related to COPD [35]. Interestingly, since one result of a study clearly showed that cigarette smoke-related changes in DNA methylation is reversible after quitting smoking, it signifies that DNA methylation might be a useful biomarker for COPD [43].

Smoking-related differentially methylated genes in peripheral blood cells are known to result in the development of COPD and gradual decline in lung function [44,45]. Even though there is heterogeneity in the findings obtained from various studies, some epigenetic players have been identified so far. For instance, hypomethylation in *SERPINA1* (*Serpin family A member 1*) gene at two CpG sites (cg02181506 and cg24621042 on chromosome 14) was linked to the COPD and smoker group, as well as *AHRR* (*Aryl-hydrocarbon receptor repressor*) gene hypomethylation in intron 3 (cg21161138) [43,46]. In contrast with this, *GABRB1* (*Gamma-aminobutyric acid type a receptor beta-1 subunit*) (cg15393297), *NOS1AP* (*Nitric oxide synthase 1 adaptor protein*) (cg26663636), and *TNFAIP2* (*TNF-alpha-induced protein 2*) (*cg18620571*) genes were reported to be hypermethylated in gene bodies of smokers as well as COPD when compared with a healthy group [47]. *SULF2* (*Sulfatase 2*), lung cancer associate gene, has been shown to have increased total methylation status in sputum samples of ex-smokers showing lasting symptoms of COPD, which is chronic mucous hypersecretion (CMH) [48]. On the other hand, Armstrong et al. have reported that DNA methylation profile changes of *CLIP4* (*CAP-Gly domain-containing linker protein family member 4*) (cg26118047), *HSH2D* (*Hematopoietic SH2 domain-containing*) in 5′-UTR region and *SNX10* (*Sorting nexin 10*) gene in 5′-UTR region are associated with metabolic differences in lung macrophages [49]. In another study, hypermethylation of *mtTFA* (*Mitochondrial transcription factor A*) promoter is demonstrated to be triggered via elevated cigarette smoke and to result in COPD development [50].

Interestingly, the researchers point out that there is a strong correlation between aberrant DNA methylation signals and overexpression of DNA methyltransferases (DNMT1, DNMT3a, and DNMT3b). Liu et al. have reported that after exposure to cigarette smoke condensates for up to 9 months, DNMT3b expression significantly upregulated in respiratory epithelial cells, including small airway epithelial cells and HBECs (human bronchial epithelial cells), while DNMT1 is reduced [51]. The DNA methylation pattern of transcription factors plays a critical role in goblet cell differentiation as well. In this context, Song et al. identified the hypomethylation of *FOXA2* (*Forkhead box protein a2*) and *SPDEF* (*SAM-pointed domain-containing ETS-like factor*) promoters, which are involved in inhibition of goblet cell differentiation and mucus production, respectively [52]. In a COPD small airway study group, Vucic et al. identified hypermethylation of genes involved in metabolism, glutathione S-transferases such as *GSTT1, GSTM1, GSTP1*) and cholinergic receptors such as *CHRNB1, CHRNB2, CHRND* [35].

In addition, a great number of genes encoding ECM proteins were differentially methylated in airway lung fibroblasts of COPD patients. Clifford et al. reported that several genes including WNT3a (WNT family member 3a), *TMEM44* (*Transmembrane protein 44*), and *HLA-DP1* (*Major histocompatibility complex DR beta 5*) were differentially methylated among 652 loci [53]. There are various studies indicating a biological link between COPD and lung cancer. One study among them demonstrated that hypermethylation of *WIF-1* (*WNT inhibitory factor 1*) and *IL-12Rbeta2* (*Interleukin 12 receptor, beta 2 subunit*) promoters commonly arise during the transition of COPD and add the risk for lung cancer development [54]. In addition to these findings, air pollution is also shown to alter DNA methylation. For instance, *SYTL2* (*Synaptotagmin-like 2*) (cg11691844), *WDR46* (*WD repeat domain 46*) (cg05454562), *AKNA* (*AT-Hook transcription factor*) (cg13999433) and *NEGR1* (*neuronal growth regulator 1*) (cg07721244) are differently methylated in their CpG sites due to air pollution [37,55]. Lastly, hypomethylation of *HDAC6* (*H**istone deacetylase 6*) is reported, which leads to HDAC activity and causes epithelial dysfunction [56].

Overall, these findings suggest that altered DNA methylation is involved in cigarette smoke related to COPD pathogenesis, and further investigations need to be conducted for a better understanding the underlying mechanisms.

### 2.3. Histone Modifications in Lung Fibrosis

Aside from DNA methylation, post-translational modification (PTM) of histones plays an important role in most biological processes that impact gene expression by changing chromatin structure. This process is often linked to the development, differentiation, and pathogenesis of many diseases and disorders, including pulmonary diseases. The two most common histone modifications are namely methylation and acetylation. Methylation of histones can associate with either transcriptional activation such as H3K4 [57] and H3K36 [58] or transcriptional repression on H3K9 [59], H4K20 [60], and H3K27 [61,62]. Histone methylation is regulated by a dynamic interplay between two sets of enzymes: histone methyltransferases (HMTs) and histone demethylases (HDMs). Histone methyltransferases (HMTs) add a methyl group on the side chains of lysines and arginines of the H3 and H4 histones as well as non-histone proteins. While lysines are mentioned to be mono-, di-, or tri-methylated, arginines on the other hand are suggested to be mono-, symmetrically or asymmetrically di-methylated. In contrast, histone dimethyltransferases (HDM) are capable of removing methyl groups from both histones and other proteins. The disruption of a balance between the opposing activities of HDMs and HMTs contributes to the developmental defects and tumorigenesis observed in various organs [63].

The histone acetylation signature of a cell plays an important role in the modulation of chromatin structure and gene expression. This dynamic process is regulated by the balance between histone acetyltransferase (HATs) and histone deacetylase (HDAC) activities. Histone acetyltransferases (HATs) acetylate lysine amino acid on histone tails, which promotes open chromatin structure and hence leads to increased gene expression. Conversely, histone deacetylases (HDACs) remove acetyl groups from histone tails. This causes the histones to wrap the DNA more tightly, making it less permissive for transcription factors to bind to it, which results in decreased gene expression and transcriptional repression. Among HATs, p300 is perhaps the most widely studied protein that is associated with the transcriptional activation of numerous genes in response to cellular signaling. The increased activity and expression of p300 were shown to associate with different diseases, including pulmonary fibrosis [64] and acute respiratory distress syndrome [65], and reported in different types of cancers [65,66,67,68]. Moreover, the genetic deficiency and pharmacological inhibition of p300 were shown to abrogate pulmonary fibrosis both in vitro and in vivo [64,69].

The role of HATs is opposed by HDACs. In humans, there are 18 HDACs that are grouped into five different classes according to phylogenetic and sequence homology: class I, class IIa, class IIb, class IV HDACs, which are zinc-dependent, and class III or sirtuins [70]. Class I HDACs (including HDAC 1, 2, 3, and 8) contain a deacetylase domain and are primarily localized in the nucleus of cells. Members of this class are ubiquitously expressed throughout various developmental stages and different cell types. Class II HDACs on the other hand have tissue- and stage-specific expression and can shuttle between cytoplasm and nucleus in response to various regulatory signals [71,72]. Members of Class II HDACs are divided into two subclasses: subclass IIa includes HDAC4, 5, 7, and 9 and subclass IIb consists of HDAC6 and 10 based on their primary structure. Class IIa HDACs consists of a large N-terminal regulatory domain that is required for protein–protein interactions and has important roles in the recruitment of various cofactors in addition to a C-terminal catalytic domain. Class III HDACs or sirtuins (including all sirtuins from 1 to 7) and class IV (HDAC11) display enzymatic activity to a certain extent [73].

HDACs are involved in the deacetylation of not only chromatin proteins, which can lead to altered gene-transcription regulation, but also of various non-histone proteins, regulating their function, stability, cellular localization, and/or protein–protein interaction [74,75]. Moreover, HDACs are known to modulate the expression of a large number of genes in different ways. For instance, they can form corepressor complexes with the nuclear receptor in the absence of a ligand. Studies show that HDACs can also directly interact with various transcription factors such as oncosuppressor protein p53, Stat3, GATA2, E2f, HIF1α, the retinoblastoma protein, NF-κB, Hsp90, and TFIIE [75,76,77]. Many HDACs can associate with multiprotein corepressor complexes, such as the transcriptional corepressors N-CoR, mSin3, and SMRT. These multiprotein complexes can interact with other proteins such as nuclear receptors, transcription factors, and other epigenetic gene modifiers, such as histone methyltransferases (HMTs), DNA methyltransferases (DNMTs), and methyl-CpG-binding proteins (MBDs) to regulate gene expression [78].

HDACs are known to regulate different cellular processes and are closely linked with tumorigenic features such as proliferation, metastasis, differentiation, and apoptosis-resistance phenotype of cells. Moreover, the aberrant expression or activity of HDACs is usually associated with human cancers and poor prognosis [79,80,81,82,83,84,85,86]. Aberrant HDAC activities are also observed in fibrotic diseases including renal [87], cardiac [88], liver [89], and pulmonary fibrosis [90]. In each study the specific mechanism of HDACs is different; however, based on the studies, the evidence suggests that HDACs usually trigger fibrogenesis in various ways and that increased expression of HDACs stimulates fibroblast to myofibroblasts trans-differentiation [91,92]. A wide-scale study examining the expression of HDACs in IPF demonstrated that nearly all class I and II HDAC enzymes are upregulated in IPF lung tissue. Moreover, upregulation of HDACs was predominantly observed in myofibroblasts of fibrotic foci region as well as in abnormal bronchiolar basal cells in areas of bronchiolization in IPF lungs [93]. Furthermore, they showed that HDACi such as LBH589 may be useful for the treatment of IPF, interfering with fibroblast to myofibroblasts differentiation, and more importantly leading to the downregulation of ACTA2 and ECM genes such as COL1A1, COL3A1, and FN in primary IPF fibroblasts.

### 2.4. Histone Modifications in COPD

Several studies demonstrated that the decrease in HDAC activity results in upregulated transcription status of pro-inflammatory cytokine genes, and in the case of HDAC overexpression, lung neoplasia is promoted as the transcription of tumor suppressor genes is decreased. Since histone acetylation and deacetylation are critical regulators of chromatin structure and gene expression, imbalance in between promotes susceptibility to the development lung diseases, including COPD and carcinogenesis in smokers [94,95]. Almost two decades ago, the role of HATs (CBP, GCN5, p300, and PCAF) was suggested in COPD pathogenesis, and it is demonstrated that HAT inhibitors targeting these molecules could be used in clinical applications as a therapeutic aspect [96]. In another study, Adenuga et al. identified that HDAC2 expression and enzyme activity in specimens of COPD patients were significantly reduced, which is thus correlated with excessive inflammation [97]. Decreased HDAC2 activity is associated with glucocorticoid resistance and elevated oxidative stress as well as with pro-inflammatory mediators in smoke-induced COPD [98]. Under normal conditions, HDAC2 plays a role in T cell differentiation and IL-17 production. Lai et al. reported that HDAC2 activity inhibits airway remodeling in lung tissue samples of COPD patients by repressing IL17A production. Hereby, this study confirmed that there is a strong correlation between the dysregulation of HDAC2/IL17A axis and bronchial wall thickening as well as collagen deposition [99]. A similar study using curcumin, an anti-inflammatory drug, in order to modulate HDAC2 expression indicated that inflammation is resolved and corticosteroid resistance is recruited in AECII (alveolar epithelial type II cells) in a COPD rat model [100]. One of the interesting effects of HDAC dysregulation is that it is capable of stimulating pathological responses. Ding et al. noted that the expressions of HDAC1 and HDAC2 were upregulated in mice exposed to cigarette smoke and led to skeletal muscle atrophy. The group also reported that the use of TSA (Trichostatin A), a global HDAC inhibitor, reserved skeletal muscle atrophy by inhibiting HDAC1 and 2 activities [101]. There is another study supporting the evidence that selective HDAC1-3 inhibitor MS-275 contributed to anti-inflammatory effect in a cigarette smoke mice model via stimulating IL-10 production [102].

One of the HDAC class enzymes, SIRT1 (silent inflammation regulator 1), is known to play role in inflammation, aging, cell senescence, and emphysema formation in COPD [103]. Ma et al. demonstrated that upregulated SIRT1 expression controlled NF-κB acetylation and repressed inflammatory responses through erythromycin [104]. Vincenzo et al. also noted that deficiency of SIRT1–FOXO3 interaction resulted in aberrant inflammatory responses in bronchial epithelial cells [105].

Apart from histone acetylation/deacetylation, histone methylation in COPD can regulate gene transcription based on the modified lysine and arginine residues, where the formation could be mono-, di-, or tri-methylated [106]. H3K4me3 is an important histone marker, linked with transcription activation [107]. Similar to DNA methylation, the H3K4me3 epigenetic marker plays a critical role in mammalian development [108]. Alteration of this marker is already associated with cancer and other diseases [109,110,111]. Since the histone methylation concept in COPD is a more complicated process than acetylation/deacetylation, this research field is still narrow. However, Yildirim et al. indicated that elevated mRNA and protein expression of PRMT2, a protein arginase methyltransferase, was a stimulus for COPD development in a hypoxia-exposed mice model [112]. Moreover, Andresen et al. noted that increased mRNA level of *DEFB1* (*Beta defensin 1*) is correlated with H3K4me3 in progression of COPD [113]. Recently, He et al. demonstrated that PRMT6, a protein arginase methyltransferase induced by H3K4me3, regulates NF-κB activation negatively [114]. Taken together, histone modifications seem to have a crucial role in the development and progression of COPD and IPF.

### 2.5. MicroRNAs in IPF

MicroRNAs (miRNAs) are a class of small non-coding RNAs and key epigenetic regulators that can bind to the 3′ untranslated region of mRNA targets and mediate their degradation. Several studies have mentioned the role of miRNA in lung development and pulmonary diseases, such as asthma, cystic fibrosis, chronic obstructive pulmonary disease, and pulmonary artery hypertension [115]. Moreover, miRNAs have been associated with almost every stage in the pathogenesis of IPF including lung epithelial repair; epithelial–mesenchymal transition (EMT) such as let-7d, miR-200, miR-26a, and miR-375; activation of fibroblast and their trans differentiation to myofibroblasts such as miR-21, miR-155, miR-26a, miR-27a-3p, miR-9-5p; AECII cell senescence; and regulation of collagen production such as miR-320a [116,117,118]. MicroRNAs are both upregulated and downregulated during the pathogenesis of IPF and were shown to have both pro-fibrotic and anti-fibrotic roles in the pathogenesis of the disease [119]. Interestingly a study identified 47 significantly differentially expressed serum miRNAs from IPF patients compared with healthy controls including 21 upregulated miRNAs and 26 downregulated miRNAs [120]. Moreover, these miRNAs were shown to regulate important biological processes that are known in the pathogenesis of the disease, such as TGF-β signaling, MAPK signaling, PI3K–Akt signaling, Wnt signaling, HIF-1 signaling, Jak–STAT signaling, Notch signaling pathway, and regulation of actin cytoskeleton. Finally, as the application of miRNAs as novel diagnostic and therapeutic tools in lung diseases has become more and more attractive, it still faces significant challenges, including non-specific targets, specific delivery to the targeted cells, activation of the innate and adaptive immune responses, and possible cytotoxicity.

### 2.6. MicroRNAs in COPD

As a research field, microRNAs (miRNAs) in COPD have been an exciting topic to explore for a better understanding of COPD pathogenesis so far. A growing number of studies indicate that miRNAs are involved in many pathways, such as tissue repair and inflammation, that have important roles in emphysema and COPD pathogenesis. The dysregulated miRNA expression has been observed in lung samples of COPD patients when compared with smokers as a control group [121]. High throughput platforms have enabled the investigation of 70 differentially expressed miRNAs in lung tissue samples of smokers with or without airflow limitations. According to this research study, miR-422a, miR-923, and miR-937 significantly downregulated in COPD patients, while miR-144, miR-223, and miR-1274a upregulated [121]. In another study, miR-203 was found to be downregulated in lung tissue while being upregulated in blood samples of COPD patients when compared with non-smokers [122]. Further, 56 differentially expressed miRNAs were identified in patient blood samples of smoke-induced COPD. Furthermore, it was demonstrated that the expression level of miR-26a-5p, miR-149-3p, miR-451b, and miR-3202 were significantly downregulated in smokers with or without COPD when compared with the non-smoker group. In addition, miR-149-3p was highlighted to control the NF- κB pathway by regulating *TLR4* (*Toll like receptor 4*) response in THP-1 cells in a murine monocytic cell line [123]. In another study, TLR4 mRNA was shown to be regulated by miR-1236, which leads to risk of development of VAP (ventilator-associated pneumonia) in COPD [124]. In another study, Pottelberge et al. noted that miR-125b and let-7c in sputum supernatant were reduced in COPD patients when compared with the non-smoker control group. Interestingly, the group discovered that let-7c was inversely linked with its target TNFRII (tumor necrosis factor receptor type II) in the sputum of patients [125]. Some studies have demonstrated that the Notch signaling pathway has a critical role in COPD since *Notch3* (*Notch receptor 3*) expression was shown to be inhibited by miR-206 in human pulmonary microvascular endothelial cells of smokers with COPD [126]. Long et al. revealed that miR-34a inhibits *Notch1* (*Notch receptor 1*) gene expression in endothelial cells exposed to chronic smoke [127].

A number of papers noted that there is a strong correlation between COPD and lung cancer. Currently, COPD is suggested to be a driver of lung cancer since many lung cancer-related death cases were associated with smoking [128]. Chronic exposure to cigarette smoke is known to cause oxidative stress through free radicals and epithelial–mesenchymal transition (EMT), these two changes of which are highly present in COPD as well as in lung cancer pathogenesis. For instance, miR-200 is found to suppress tumor progression via binding to *ZEB1* (*Zinc finger E-box-binding homeobox 1*) and *E-cadherin*, eventually inhibiting the EMT process. The regulation of miR-200 expression is also linked with metastasis in lung cancer [129,130].

In addition, several miRNAs are identified to be involved in emphysema pathogenesis. Christenson et al. demonstrated alteration of 63 differentially expressed miRNAs such as miR-181d, miR-30c, and miR-638 in regional emphysema of COPD patients through comparing distinct locations of varying grades of emphysema. These miRNAs target corresponding genes associated with oxidative stress and progressive aging in emphysema [131]. Moreover, Savarimuthu et al. reported that miR-34c targets *SERPINE1*, a protease inhibitor, in emphysema via altering protease/anti-protease balance [132].

Strikingly, there is another emerging research field in COPD pathogenesis related to miRNAs, which is their transportation within EVs (extracellular vesicles). EVs are lipid bilayer structures released by all cell types and present in nearly all body fluids including BALF (broncho alveolar lavage fluid). EVs can be classified based on their size and origins. According to these criteria, exosomes are >100 nm while microvesicles are larger than 100 nm. EVs are responsible for cell-to-cell communication and cell homeostasis by transporting nucleic acids including miRNAs, proteins, lipids, and other bioactive molecules [133]. Since EVs are shown to contain epigenetic marks such as DNMTs and HDACs, it is most likely that they are involved in epigenetic regulation of lung diseases as well [134,135]. The exosomes released from BECs (broncho epithelial cells) exposed to chronic exposure are shown to bear extracellular matrix-associated CCN-1 (Cellular communication factor 1) protein, which induces MMP-1 (Matrix metalloproteinase 1) secretion leading to emphysema formation eventually [136]. Similarly, AMs (alveolar macrophages) release EV and MV when exposed to chronic cigarette smoke in order to upregulate pro-inflammatory mediators such as IL-8 (Interleukin 8), ICAM-1 (Intercellular adhesion molecule 1), and MCP-1 (Monocyte chemoattractant protein 1) in AECs [137]. Another study reported that EVs bearing miR-210 derived from COPD patients were found to silence *ATG7* (*Autophagy related 7*), autophagy-related factor, during fibroblast to myofibroblast differentiation [138]. Finally, MVs released from endothelial cells after cigarette smoke were identified to be enriched by miR-125a, miR-126, miR-191, and let-7d. Once these miRNAs were transported as cargo by MVs, they influenced efferocytosis of recipient macrophages [139]. These results suggest that EVs have a crucial regulatory role in COPD. Thus, transportation of small RNAs through EVs might be a novel mechanism that could result in spreading altered bioactive molecules and epigenetic marks from a single cell to another. Further investigations are needed to elucidate the underlying mechanism in COPD.

## 3. Conventional Medications and New Strategies for IPF and COPD

There is no effective cure for IPF and COPD so far. However, medication and other treatment options can help improve patients’ quality of life. A number of comorbidities are often linked to IPF, including lung cancer, pulmonary hypertension, emphysema, depression, cardiovascular disease, thrombosis, acute respiratory distress syndrome (ARDS), and respiratory failure, which causes further difficulties or delays in diagnosing and treating IPF [140,141,142,143,144,145,146,147]. Currently, there are two drugs approved by the FDA (U.S. Food and Drug Administartion) for the treatment of idiopathic pulmonary fibrosis (IPF). These include nintedanib and pirfenidone. However, as these agents do not show curative effects [148], new therapeutic approaches for patients are required. On the other hand, anti-inflammatory inhaled corticosteroids in combination with bronchodilators have been in use for patients with COPD [11,23] (Figure 3). However, large numbers of COPD patients are resistant to corticosteroids, as mentioned earlier.

To improve the current therapeutic options, a considerable shift in efforts towards the identification of signaling mechanisms involved in the pathogenesis of IPF and COPD was observed. Since then, perturbations of several molecular mechanisms, including pathways involving growth factors, cytokines, metabolic signaling, and transcription factors and epigenome that may underlie the pathogenesis of the disease, have been described. Recently, HDACs have gained increasing attention, as HDAC-inhibiting compounds were shown to correct abnormalities in various cell process including proliferation, migration, vascularization, and death. Although many researchers have shown the valuable effects of HDAC inhibitors against multiple human diseases, they can simultaneously induce acetylation of histones as well as non-histone proteins involved in regulation of gene expression and in various cellular pathways. Thus, they can contribute to toxic and fatal side effects. Targeting HDACs in an isoform-specific manner could achieve enhanced clinical utility by reducing or eliminating the serious side effects associated with current first-generation non-selective HDAC inhibitors [149,150]. Similarly, conventional DNMT inhibitors (DNMTIs) lack specificity for gene(s) of interest and were shown to induce the demethylation of not only tumor suppressor genes but also oncogenes [151]. Thus, they are not capable of specifically reversing the aberrant DNA methylation regulation involved in the pathology of lung diseases. The current therapies with epigenetic modifying drugs (epi-therapies) to treat pulmonary diseases are still in their infancy. However, comprehensive and integrated studies are required to delineate specific epigenetic events and their consequences in each cell type and each lung disease.

Recently, a new therapeutic era has emerged in the COPD field: so-called epigenetic editing. Epigenetic editing is a valuable tool for writing or erasing epigenetic modifications at the target gene of interest in order to modulate its expression. The epigenetic editing tool is a combination of an epigenetic effector domain with a DNA-binding domain coupled to it. The epigenetic effector domain is a catalytic domain of epigenetic enzyme that alters the epigenetic status of the target locus via writing or erasing the modifications. The DNA-binding domain coupled to an epigenetic effector domain recognizes the specific sequence at the desired gene. Here, the ultimate goal is to upregulate or downregulate the expression of the desired genes [152]. So far, numerous epigenetic modifier systems have been employed to alter gene expression via introducing epigenetic editors to the cells with various transfer methods. However, lung delivery of epigenetic modifiers has its own limitations in terms of successful applications. Since lung has both immune and physical barriers to keep the environment pathogen-free, delivering these epigenetic modifiers to the cells is a major challenge. The mentioned barriers consist of tight junctions among epithelial cells, alveolar macrophages, which are responsible for taking up and clearance of infectious and toxic particles, and lastly mucociliary escalator [153]. The other challenge is the broad actions of these epigenetic editors, as they mostly have multiple targets, which consequently might cause deleterious side effects. Targeting and modifying gene expression at the DNA level is more advantageous when compared with targeting RNA or protein. In order to inhibit RNA or protein effectively, continuous administration of epigenetic tools is necessary, while targeting DNA is to silence the source of expression directly. In addition, potential splicing isoforms of RNA and translated distinct proteins could be another compelling point to consider in applications for lung diseases. In the matter of upregulation of gene expression at the DNA level, this leads to increased gene expression due to all possible isoforms in natural ratios. In fact, ectopic cDNA delivery leads to over-expression of only the desired isoform of the gene of interest. For this purpose, ATFs (artificial transcription factors) have been used to modify genes at the DNA level. ATFs include DNA-binding domains coupled to transcription effector domains, which can be a transcription activator domain such as *VP16* fused to *VP64* (*Herpes simplex virus protein VP16 and its tetramer VP64*) or transcription repressor domains such as *SKD* (*Super KRAB domain*). TFO (triplex forming helix), ZFPs (zinc finger proteins), TALEs (transcription activator-like domains), and CRISPRs (clustered regularly interspaced short palindromic repeats) have been developed to target and modify the desired DNA sequences [154]. Up to now, a limited number of research studies have been reported regarding DNA targeting system in COPD. In order to study the roles of genes linked to inflammation, surfactant production, and epithelial cell senescence in the pathogenesis of COPD, knock-out cell and animal lines have been created by CRISPR-editing. For instance, Chu et al. demonstrated the role of *MUC18* was found to be pro-inflammatory in CRISPR-edited primary human airway epithelial cells [155]. In another study, Zhang et al. studied pulmonary surfactant synthesis in CRISPR-mediated miRNA-26a-1/miRNA 26a-2 double knockout mice [156]. Moreover, *GDF15* (*Growth differentiation factor 15*) production was shown to be involved in cigarette smoke-induced epithelial cell senescence by a CRISPR-mediated knock-out strategy [157]. The airway mucus hypersecretion was targeted by epigenetic editing since it contributes to COPD pathogenesis. Song et al. revealed that ZFPs and CRISPR-mediated gene silencing of *SPDEF* (*SAM-pointed domain-containing Ets-like factor*) reduced excessive mucus secretion in lung epithelial cells, suggesting a therapeutic strategy for COPD patients [158]. The researchers utilized a system including the addition of histone and DNA methylation as well as transcription repressors to silence the promoter of *SPDEF* gene.

Gene transfer with non-viral and viral approaches has been investigated broadly in the respiratory diseases. Particularly, viral vectors are reported to be quite efficient, even in the clinical practice setting since they have a natural tropism to the respiratory system [159,160]. Alton et al. noted that F/HN-pseudo typed SIV vector was shown to produce gene expression in the lungs during their lifetime. Based on these promising data, the group built up the rSIV.F/HN vector into a first-in-man CF (cystic fibrosis) clinical trial [161]. However, there are a limited number of studies reported in the COPD aspect. For the non-viral approaches, commercially available liposomes such as lipofectamine, PEG (polyethylene glycol), GL67A) have been employed as a strategy for delivering a gene into a target cell, as well as lipid nanoparticles. Mastorakos and coworkers produced highly stable and non-toxic polymers, so called BPAEs (β-amino esters), in order to treat the mucosal layer above the respiratory epithelium [162]. In other studies, PBAE-MMPs (PBAE-based mucus-penetrating DNA nanoparticles) were highlighted to have a certain extent of success regarding transfection efficiency and long-term effect after repetitive administration [163]. Additionally, Mahiny et al. provided a successful example of gene editing in surfactant B (SP-B)-deficient mice via using nuclease-encoding mRNA coupled with chitosan-coated nanoparticles, a safe polymer [164]. Chitosan nanoparticles were also reported to be used as a protein delivery system [165]. Since there is a strong correlation between distinct miRNA signature patterns and disease progression in COPD, miRNA profiling has been studied widely. In one study, miR-146a incorporated with NCMPs (nanocomposite microparticles) was successfully delivered to reduce *IRAK1* (*IL-1 receptor-associated kinase*) and *TRAF6* (*TNF receptor-associated factor 6*) gene expressions for COPD treatment purposes [166]. Cadmium, one of the components of cigarette smoke, is known to trigger inflammation and contribute to COPD pathogenesis. Interestingly, miR-181a-2-3p expression was attenuated while inflammatory response was elevated in cadmium (Cd)-treated human bronchial epithelial cells [167]. Additionally, miR-197 expression was correlated with the contractile phenotype in SMCs (smooth muscle cells). The researchers showed that miR-197 inhibition provoked migration while preventing the acquisition of SMC contractile markers [168]. Clearly, these studies point out that some miRNAs might be considered as biomarkers and a crucial target for a therapeutic approach in COPD. For instance, downregulation of miR-3177-3p and miR-183-5p in peripheral leukocytes is an important biomarker for COPD [169]. Due to these promising studies, epigenetic editing opens a novel and exciting therapeutic avenue in lung disease therapy.

## Figures and Tables

**Figure 1 ijms-23-00546-f001:**
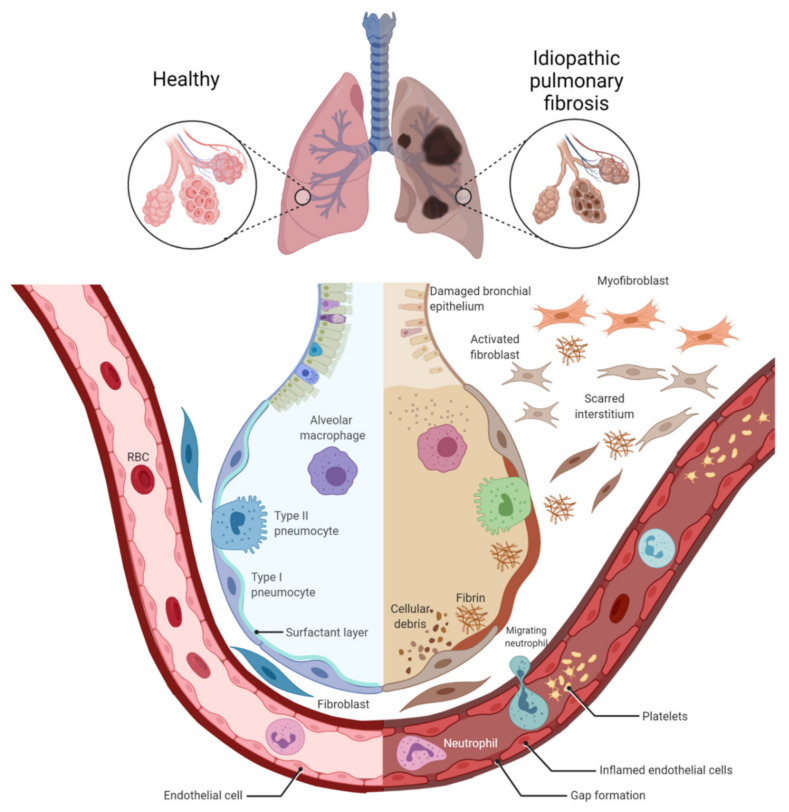
Alveolar damage, dysfunction of inflammatory cells, and scarred interstitium in idiopathic pulmonary fibrosis. Idiopathic pulmonary fibrosis results in dilatation of the bronchi, inflammation, alveolar remodeling, and parenchymal fibrosis (fibroblast–myofibroblast differentiation), which results in impaired gas exchange. Figure created using BioRENDER.com. Accessed on 5 December 2021.

**Figure 2 ijms-23-00546-f002:**
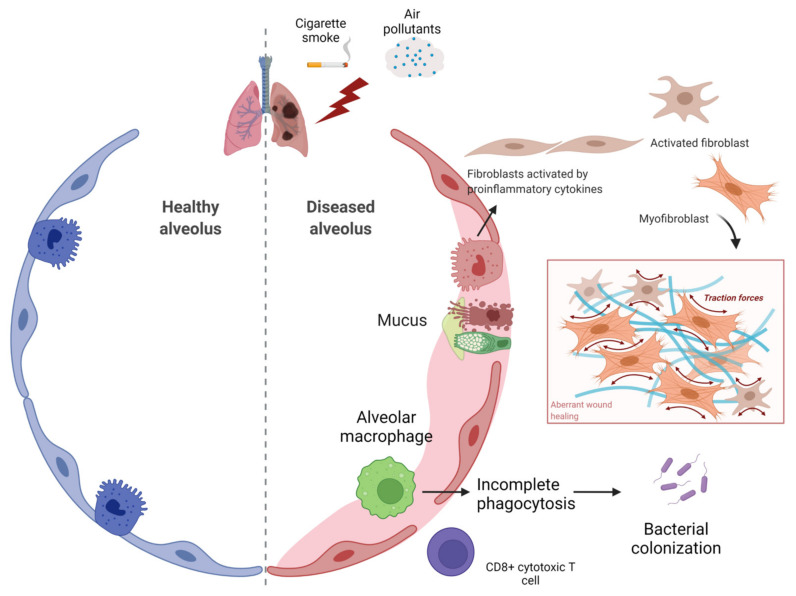
Airway obstruction and inflammatory cells involved in COPD. In chronic obstructive pulmonary disease (COPD), decline in lung function and airflow obstruction due to cigarette smoke and various air pollutants results in disruption of alveolar wall attachments as a result of emphysema formation and airway luminal occlusion by mucus hypersecretion. Both inflammation and fibrosis lead to narrowing of small airways via thickening of the bronchiolar wall. Figure created using BioRENDER.com. Accessed on 18 November 2021.

**Figure 3 ijms-23-00546-f003:**
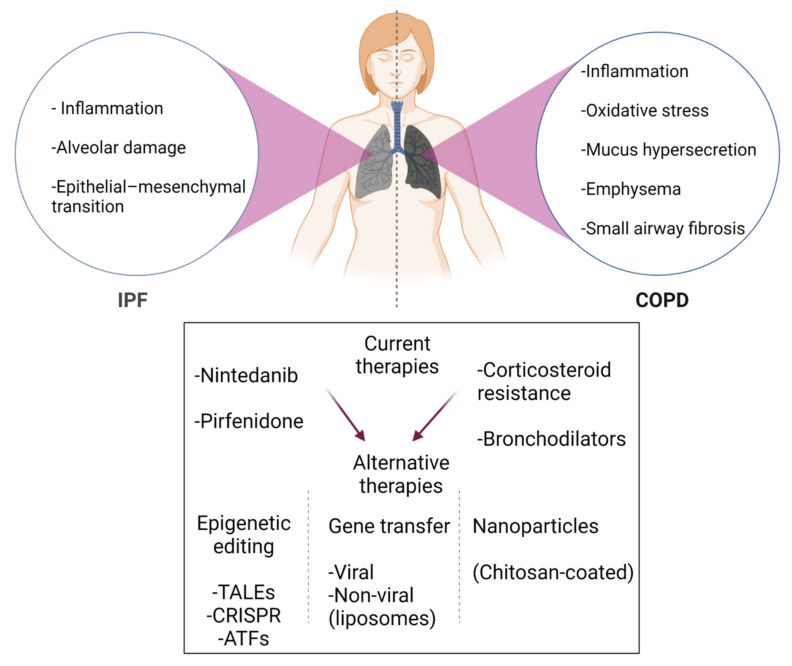
Brief schematic summary of current medications and new strategies for IPF and COPD treatment. Since current drugs and therapies do not show curative effects, new approaches for patients (depicted by red arrows) are required. In addition, large numbers of COPD patients are resistant to corticosteroids. To improve the current therapeutic options in IPF and COPD, a considerable shift in efforts towards epigenetic editing, gene transfer, and nanoparticles in the research field. Figure created using BioRENDER.com. Accessed on 18 November 2021.

## Data Availability

Not applicable.

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
