# Peer review of "Epigenetic Mechanisms in Parenchymal Lung Diseases: Bystanders or Therapeutic Targets?"

_ijms, 2022, doi:10.3390/ijms23010546_

Round 1
Reviewer 1 Report
The authors proposed an article focused on the different epigenetics mechanisms that may participate in the development of idiopathic pulmonary fibrosis and chronic obstructive pulmonary disease. This work offers an exhaustive view of the main epigenetic modifications that occur in these pathologies. In the second part of the manuscript, they developed a very interesting paragraph on the therapeutics leverage that could be used to correct the aberrant epigenetic landscape observed in these pathologies. I suggest some minor comments to the manuscript:
General comments:
- please justify the text all along the document.
- for the methylation part (2.1 and 2.2), please describe the localization of the methylation change of the genes you cite (promoter, gene body, enhancer).
- authors Contribution, Funding, and Institutional Review Board Statement have not been completed.
Specific comments
Line 19-20: this part of the sentence is not clear for me: “which cultured ex vivo characterized and showed to maintain the disease phenotype”.
Line 33: please add information on the time course of this pathology (survival, death rate,…)
Line 47: please correct: “(Fig. )1”.
Line 48: I suggest to moderate this sentence by changing “responsible for” by “participating to”.
Line 50: the end of sentence is missing.
Line 71-72: You refer to WHO records from 2019 in this sentence, please find a most recent publication to justify “the prevalence continues to increase by 2021”.
Lines 82-85: the causes and consequences are not differentiated. For example in this sentence “apoptosis and inflammation” are molecular mechanisms that may participate in the phenotypes changes such as “remodeling of blood vessels as well as extracellular matrix”.
Lines 85-86: “Inflammation itself is a major influencer of the ROS response in the cells” please add a citation.
Line 121-122: DNA hypermethylation is associated with an altered level of gene expression, not to a systematic suppression of its expression. Indeed, It has been shown that over-methylation could be associated with an increased expression of some genes (see D. Anastasiadi et al. 2018 in Epigenetics & Chromatin).
Line 124: transfert the information on COPD methylation “and COPD (33, 34)” in the “2.2. DNA methylation in COPD” paragraph.
Line 144-145: Changing “expression” by a synonym (i.e. production) may avoid confusion with the previous sentence that describes no change in the expression of DNMT3A.
Line 152: please correct “d pendent” by “dependent”.
Line 196: precise that the study of Liu et al is focused on chronic CSC exposure (up to 9 months)
Line 284-285: please add a reference.
L307-318: It could be interesting to discuss the opposing results obtained by Adenuga (102) on HDAC2 expression and activity when compared to other authors (103).
Line 351: please add a short paragraph introducing the mechanisms of action of the miRNAs as you did for others epigenetics mechanisms.
L355-356: please add a reference.
Reviewer 2 Report
Excellent and outstanding review.
I would only add a chapter about epigenetic in (especially severe) asthma, since it is very often associated to airway remodeling and there are many studies in literature about this.
